# The Roles of the *PSEUDO-RESPONSE REGULATORs* in Circadian Clock and Flowering Time in *Medicago truncatula*

**DOI:** 10.3390/ijms242316834

**Published:** 2023-11-28

**Authors:** Xiao Wang, Juanjuan Zhang, Xiu Liu, Yiming Kong, Lu Han

**Affiliations:** 1The Key Laboratory of Plant Development and Environmental Adaptation Biology, Ministry of Education, School of Life Sciences, Shandong University, Qingdao 266237, China; sdnuwx@126.com (X.W.); juanjuanzhang1103@163.com (J.Z.); lx18726087262@163.com (X.L.); lck7love@163.com (Y.K.); 2College of Life Sciences, Shandong Normal University, Jinan 250014, China

**Keywords:** *Medicago truncatula*, MtPRR, circadian clock, flowering time, heterodimer, MtTPL/MtTPR

## Abstract

*PSEUDO-RESPONSE REGULATORs* (*PRRs*) play key roles in the circadian rhythms and flowering in plants. Here, we identified the four members of the *PRR* family in *Medicago truncatula*, including *MtPRR9a*, *MtPRR9b*, *MtPRR7* and *MtPRR5*, and isolated their *Tnt1* retrotransposon-tagged mutants. They were expressed in different organs and were nuclear-localized. The four *MtPRRs* genes played important roles in normal clock rhythmicity maintenance by negatively regulating the expression of *MtGI* and *MtLHY*. Surprisingly, the four *MtPRRs* functioned redundantly in regulating flowering time under long-day conditions, and the quadruple mutant flowered earlier. Moreover, MtPRR can recruit the MtTPL/MtTPR corepressors and the other MtPRRs to form heterodimers to constitute the core mechanism of the circadian oscillator.

## 1. Introduction

Plants have evolved an internal circadian biological clock system with a period of about 24-h to regulate important activities at the right time of day [1,2]. Most studies on the plant circadian clock have been investigated in *Arabidopsis thaliana* and included many processes, such as flowering time regulation, hormone synthesis, signal transduction, stress response pathways, and interactions between the plant and pathogens [3,4,5,6]. The traditional circadian clock consists of three major components: the central oscillator, clock input pathways, and clock output pathways. The central oscillator is made up of several highly interconnected transcriptional and post-transcriptional feedback loops [1,7,8,9]. The *CIRCADIAN CLOCK ASSOCIATED1* (*CCA1*), *LATE ELONGATED HYPOCOTYL* (*LHY*), *LUX ARRHYTHMO* (*LUX*), *EARLY FLOWERING 3* (*ELF3*), *EARLY FLOWERING 4* (*ELF4*) and *PSEUDO-RESPONSE REGULATOR* (*PRR*) genes are major members of the central oscillator [10]. *CCA1* and *LHY* function redundantly to repress the expression of *TIMING OF CAB EXPRESSION1* (*TOC1*), *LUX*, *ELF3*, and *ELF4*, but to promote the expression of *PRR9* and *PRR7*, while *PRR9* and *PRR7* could also suppress the expression of *CCA1* and *LHY* in a feedback loop in the morning [11,12,13,14]. LUX, ELF3, and ELF4 form a trimeric complex named as the evening complex (EC), which represses the expression of *GIGANTEA* (*GI*), *PRR9*, and *PRR7* in the evening [15]. 

The PRR proteins have an N-terminal pseudo-receiver (PR) domain and a C-terminal CTT (CONSTANS, CONSTANS-LIKE, TOC1) domain [16]. They are highly conserved in the circadian rhythm in *Arabidopsis thaliana* [17,18]. *PRR9*, *PRR7*, *PRR5*, *PRR3*, and *TOC1*/*PRR1* are members of the *Arabidopsis* PRR family, and these genes are regulated by the circadian clock in a temporal order [17]. Mutation in *TOC1* or *PRR5* results in a short period, while mutations in *PRR9* or *PRR7* lead to an extended period [19,20,21]. Arrhythmia is only observed in the *prr5 prr7 prr9* triple mutant in continuous light conditions, and the triple mutant also exhibits less sensitivity to photoperiodicity and photomorphogenic responses [22]. *ZEITLUPE* (*ZTL*), which encodes an F-box protein, recognizes the PR domains of TOC1 and PRR5 to promote their protein degradation, and TOC1 could form heterodimers with PRR3 and PRR5 by the PR domains to influence the protein stability of TOC1 [23,24]. These findings suggest that the PR domains are critical to protein–protein interactions. TOC1 could bind to the promoters of *CCA1* and *LHY* to suppress the gene expression, while mutation or deletion of the CCT domain in TOC1 prevents this repression, and PRR5 also binds to the *CCA1* promoter region through its CCT domain, indicating that the CCT domain of PRRs is necessary for DNA binding [16,25]. 

Flowering time regulation is extremely important in the regional climatic adaptation of elite germplasm, and floral transition determines the production of crops [26]. *Medicago truncatula*, which functions as a model plant in legume forage, is a temperate species. The flowering time of *M. truncatula* is promoted by exposure to a prolonged period of cold conditions, so it is cultivated in a relatively limited geographic range [27,28]. Exploring the genes involved in flowering time regulation will be helpful to understand the climate adaptation of *M. truncatula*, and some important genes have been reported so far. For example, *FTa1*, one of the *FLOWERING LOCUS T* (*FT*) orthologs in *M. truncatula*, is a key regulator of flowering time, and the loss of function of *FTa1* exhibits late flowering, while overexpression of *MtFTa1* accelerates flowering time [29]. A legume-specific gene in *Glycine max* was named *E1*, and the mutation of the homolog of *E1* causes a delay in flowering in *M. truncatula* [30]. In addition, *SUPPRESSOR OF OVEREXPRESSION OF CONSTANS1* (*SOC1*), *PHYTOCHROME A* and *FRUITFULL-LIKE* genes promote the flowering in *M. truncatula* [31,32,33]. But the functions of *MtPRRs* in flowering time regulation are still unknown. The *PRR* genes have been reported to regulate flowering time through the expression of *FT* in *Arabidopsis* [34]. *TOC1* was reported as a modulator of flowering time, and mutation of *TOC1* exhibits early flowering, but *toc1* enhances the late flowering phenotype of *prr5*, indicating a special function of *TOC1* in flowering time control [35,36]. PRR5 protein directly represses the expression of *CDF* (*CYCLING DOF FACTOR*) genes, which are involved in the clock output pathways and flowering time regulation [16]. Furthermore, the mutant of *prr7* shows late flowering time, while *PRR7* overexpression plants exhibit early flowering, indicating *PRR7* functions as a positive factor in floral transition [19,37]. In addition, *Photoperiod-H1*, which belongs to the *PRR* family in barley (*Hordeum vulgare*), negatively controls the expression of *HvFT*, and the mutant is late flowering [38]. On the contrary, in short-day (SD) crops, such as rice (*Oryza sativa*) and sorghum (*Sorghum bicolor*), *PRR* family genes *OsPRR37* and *SbPRR37* regulate late flowering [39,40]. Interestingly, overexpressing the *AtPRR5* in rice causes the late flowering phenotype [41]. Furthermore, *Time of Flowering 11* (*Tof11*) and *Tof12*, two homologous genes of *AtPRR3* in *Glycine max*, act via *GmLHY* to promote the transcription of E1 and delay flowering time under long-day conditions, and also improve the adaptability of soybean to high latitudes during domestication [42,43,44].

In *Medicago truncatula*, a few clock genes have been reported, including the *MtLHY* gene, which orchestrates the endogenous circadian rhythm in nodules and positively regulates the salt stress response [45,46], *MtLUX* controls the expression of clock genes, nodule formation, leaf movement, and flowering time [47]. Circadian clock genes form a complicated regulatory network, and the regulation mechanism remains unclear in *M. truncatula*. Here, we characterized the *Tnt1* retrotransposon-tagged *mtprr9a*, *9b*, *7*, and *5* mutants in *M. truncatula*. We demonstrated that the four *MtPRRs* play vital roles in maintaining clock gene expression and regulating flowering. In addition, MtPRR could form complexes with MtTPL/MtTPR corepressors and another MtPRR, indicating the post-transcription regulation of the circadian clock in *M. truncatula*.

## 2. Results

### 2.1. Identification of PRRs and Their Loss-of-Function Mutants in M. truncatula

To identify the putative orthologs of *PRRs* in *M. truncatula*, the protein sequences of PRRs in *Arabidopsis thaliana* were used as a query in BLAST searches against the protein sequence database of the *M. truncatula* genome in the National Center for Biotechnology and the Phytozome database. Based on the homology analysis, seven *M. truncatula* PRR proteins, namely MtPRR9a, MtPRR9b, MtPRR7, MtPRR5, MtPRR3, MtTOC1a, and MtTOC1b were isolated (Figure 1A). The phylogenetic analysis also showed that MtPRR9a and MtPRR9b were more closely related to MtPRR5 (PRR5/9 clade) and MtPRR3 was close to MtPRR7 (PRR3/7 clade), whereas MtTOC1a and MtTOC1b were grouped into the TOC1 clade and separated from the other members of MtPRRs (Figure 1A). Subsequently, the amino acid sequence alignments of the PRR putative orthologs were carried out, and the result showed two types of conserved domains, including the PR domain and the CCT domain (Appendix A), which are characteristic of the PRR proteins. 

To determine the functions of *MtPRRs* in *M. truncatula*, loss-of-function mutants of *MtPRR9a*, *9b*, *7*, and *5* were isolated by a reverse genetic screening in the *Tnt1* retrotransposon-tagged mutant collection of *M. truncatula*, but the mutant of *MtPRR3* failed to be isolated [48]. We focused on *MtPRR9a*, *9b*, *7*, and *5* in the subsequent study. Sequence analysis showed that a single *Tnt1* was inserted in the first exon of *MtPRR9a* and *MtPRR9b* in *mtprr9a-1* and *mtprr9b-1*, the third exon of *MtPRR7* in *mtprr7-1*, and the sixth exon of *MtPRR5* in *mtprr5-1*, respectively (Figure 1B–E). Reverse transcription PCR (RT-PCR) data showed that the transcripts of *MtPRR9a*, *9b*, *7*, and *5* were not detected in the *mtprr9a-1*, *mtprr9b-1*, *mtprr7-1*, and *mtprr5-1* mutants, indicating that the transcriptions of the four *MtPRRs* were disrupted by *Tnt1* insertion (Figure 1F–I).

### 2.2. Subcellular Localizations of MtPRR9, 7, and 5 and Expression Patterns 

To study the cellular localization of MtPRR9a, 9b, 7, and 5, the proteins of MtPRR9a, 9b, 7, and 5 were fused with green fluorescent protein (GFP) under the control of the Cauliflower mosaic virus (CaMV) 35S promoter and transformed into tobacco (*Nicotiana benthamiana*) leaf cells. Using fluorescence microscopy observation, the 35S:GFP was localized in both the cytoplasm and nucleus of epidermal cells, while the MtPRR9a, 9b, 7, and 5 fusion proteins were exclusively localized to the nucleus, which further supported their functions as transcription factors (Figure 2A).

To analyze the expression patterns of *MtPRR9a*, *9b*, *7*, and *5*, quantitative real-time PCR (qRT-PCR) was used to examine the expression levels of the four *MtPRRs* in different organs, including root, stem, leaf, flower, pod, petiole, and vegetative bud. The results showed that *MtPRR9a* and *MtPRR7* were highly expressed in leaf, petiole, and pod (Figure 2B,D), the expression levels of *MtPRR9b* were similar in all issues (Figure 2C), and *MtPRR5* showed the highest expression level in the leaf (Figure 2E).

In order to detect whether *MtPRR9a*, *9b*, *7*, and *5* are all subjected to a circadian rhythm at the level of transcription, we examined their expression patterns in plants grown in constant light conditions. Results showed that *MtPRR9a*, *9b*, *7*, and *5* transcripts varied considerably in an oscillatory manner during the given 24-h period (Figure 3A–D). Results also showed that both *MtPRR9a* and *MtPRR9b* had very similar expression patterns, and their expression levels reached their maximum after releasing into constant light at approximately 6 h (Figure 3A,B). The expression levels of *MtPRR7* and *MtPRR5* reached the peak after releasing into constant light at approximately 12 h, and the abundance of *MtPRR5* was significantly decreased at 15 h, while *MtPRR7* decreased slowly (Figure 3C,D).

### 2.3. The Expression Patterns of Genes Associated with the Circadian Clock Are Altered in mtprr Mutants

The expression patterns of *MtPRR9a*, *9b*, *7*, and *5* exhibited a diurnal and circadian rhythm, implying that *MtPRRs* may play important roles in maintaining normal clock rhythmicity. To confirm this hypothesis, two clock genes *MtLHY* and *MtGI* were analyzed in the wild type and *mtprr* single or multiple mutants under constant light conditions for two days (Figure 4). The results showed that the expression of *MtGI* maintained the robust rhythmic cycles in the *mtprr* single or multiple mutants compared with the wild type during the two days, and the expression level was higher in the *mtprr9a-1 mtprr9b-1 mtprr7-1 mtprr5-1* mutant than that in the wild type at most time points (Figure 4A,B). In addition, the expression of *MtLHY* was increased throughout the daytime, particularly at the peak time in *mtprr7-1* and *mtprr9a-1 mtprr9b-1 mtprr7-1 mtprr5-1* mutants (Figure 4C,D). These results indicated that the expression levels of circadian clock-related genes are severely compromised in the *mtprr* mutants under constant light conditions, suggesting the potential roles of *MtPRR9a*, *9b*, *7*, and *5* in regulating the clock rhythmicity. 

### 2.4. MtPRR9, 7, and 5 Regulate Flowering Time

In *Arabidopsis*, *PRR9*, *PRR7*, and *PRR5* are involved in flowering time regulation [34]. To explore whether *MtPRR9a*, *9b*, *7*, and *5* control flowering time, the flowering times of wild type and *mtprr* mutants were observed under long-day conditions. The results showed that the early flowering phenotype was exhibited in the *mtprr9a-1 mtprr9b-1 mtprr5-1 mtprr7-1* (Figure 5A,B). Under long-day conditions, the flowering time of *mtprr9a-1 mtprr9b-1 mtprr5-1 mtprr7-1* was advanced by 5–6 days, compared to that of the wild type, *mtprr9a-1*, *mtprr9b-1*, *mtprr5-1*, *mtprr7-1*, *mtprr9a-1 mtprr9b-1*, and *mtprr9a-1 mtprr9b-1 mtprr5-1* mutants (Figure 5C), indicating that the flowering time in the quadruple mutant was accelerated. 

### 2.5. MtPRR9, 7, and 5 Physically Interact with MtTPL/MtTPR Proteins

It has been shown that PRR9, PRR7, and PRR5 act as transcriptional repressors and interact with the TOPLESS/TOPLESS RELATED PROTEINs (TPL/TPRs) family to repress the transcription of *CCA1* and *LHY* [49,50]. The TPL/TPR proteins were also reported to function as corepressors in *M. truncatula* [51]. To understand the potential regulatory mechanism of MtPRR9, 7, and 5, we performed yeast two-hybrid (Y2H) experiments to examine the interaction between MtPRRs and MtTPL/MtTPRs. Results showed that MtPRR9a, MtPRR9b, MtPRR7, and MtPRR5 could interact with MtTPL in yeast cells (Figure 6A–C). Furthermore, MtPRR9a was selected to detect the interactions between MtPRR and MtTPRs, and our results showed that MtPRR9a could interact with MtTPR1-MtTPR5 in the yeast (Appendix A). These data suggest that MtPRR9, 7, and 5 may recruit the MtTPL/MtTPRs to form the complex for rhythmic regulation.

### 2.6. MtPRR9, 7, and 5 Can Form Heterodimers

It has been reported that TOC1 forms heterodimers with PRR3 and PRR5 in the regulation of TOC1 nuclear accumulation, phosphorylation, and protein stability [24,52,53]. To test the possibility of the interactions among four MtPRR proteins in *M. truncatula*, Y2H experiments were carried out. Results showed that MtPRR9a could interact with MtPRR9b, MtPRR7, and MtPRR5 in yeast cells and that MtPRR5 could also interact with MtPRR9b and MtPRR7 (Figure 6D–F). These data suggest that MtPRR9, 7, and 5 proteins can form heterodimers in the yeast. These interactions and homodimerization between MtPRR9a and MtPRR9b, MtPRR7, MtPRR5 were verified in tobacco by bimolecular fluorescence complementation (BiFC) assays. Yellow fluorescence was observed when MtPRR9a was fused to the C-terminal of YFP (YC), and MtPRR9b, MtPRR7, or MtPRR5 was fused to the N-terminal of YFP (YN) (Appendix A). Taken together, these data suggest that MtPRR9, 7, and 5 physically interact to potentially form complexes.

## 3. Discussion

The *PRR* gene family is a major component of the circadian clock, regulating various plant physiological processes, such as photomorphogenesis, maintenance of mitochondrial homeostasis, stress responses, and flowering time regulation [10]. The studies of the clock system in plants mainly focused on *Arabidopsis*; however, the circadian clock is little explored in the legume species. Here, we identified the orthologs of *PRR* in the legume model species *M. truncatula*. Our study showed that MtPRRs can be grouped into the PRR5/9 clade, PRR3/7 clade, and TOC1 clade, which is highly conserved in many species, such as *Arabidopsis*, rice, *Brassica rapa*, and rose [18,54,55]. According to the previous study, there are five members of clock *PRR* genes in *Arabidopsis thaliana*, *Sorghum bicolour*, and *Oryza sativa*, all including one *TOC1/PRR1* member gene, two genes in the *PRR5/9* clade, and two genes in the *PRR3/7* clade [41,56]. However, seven *PRR* genes were shown in the *Medicago truncatula* genome, including two genes in the *TOC1/PRR1* clade, three members in the *PRR5/9* clade, and two *PRR3/7* members. This is mainly because *PRRs* were expanded in *M. truncatula* during evolution, and the duplication event tends to happen in the TOC1/PRR1 clade and PRR5/9 clade, which is also seen in the *PRR* family of *Populus trichocarpa* and *Vitis vinifera* [56]. Like the PRR proteins in *Arabidopsis*, rice, sorghum, and other plants, the PRRs in *M. truncatula* are also nuclear-localized with an N-terminal PR domain followed by the C-terminal CCT motif, and the structure and location similarity imply their potential functional conservation in monocot and dicot species. 

The *MtPRR9a*, *9b*, *7*, and *5* genes are all under circadian control and are expressed in a sequential wave that differs from the *Arabidopsis PRRs* [19], *MtPRR9a*, and *MtPRR9b* peak firstly followed by peaks in *MtPRR7* and *MtPRR5* expression. These results suggest that there may be a change in the *PRR*-associated circadian rhythm regulatory mechanism of *Arabidopsis* and *M. truncatula*. In the *mtprr* single or multiple mutants, the expressions of *MtLHY* and *MtGI* are upregulated, implying that they may act in a linear pathway. This result is consistent with the roles of PRR5, PRR7, and PRR9 binding to the promoters of *CCA1* and *LHY* to repress their transcription in *Arabidopsis* [49]. In addition, to fully understand the relationship between *MtPRRs* and *MtLHY* or *MtGI*, the CRISPR/Cas9 genome-editing system needs to be applied to generate the *mtprr9a mtprr9b mtprr7 mtprr5 mtprr3* quintuple mutant in the future.

According to the previous study, PRRs also act by forming protein complexes [49,54,56,57]. To verify the conservation of regulatory mechanism, protein interactions among MtPRRs were performed. MtPRR9a, 9b, 7, and 5 could form dimers with other members, indicating the MtPRRs may be included in the post-transcriptional regulation of the circadian clock, such as the phosphorylation, subnuclear localization, and protein stability of MtPRRs. In *Arabidopsis*, the PRR5, 7, and 9 proteins contain a conserved EAR (ethylene-responsive element binding factor-associated amphiphilic repression) motif, specifically interacting with TPL/TPR to repress the *CCA1* and *LHY* expression [49]; similarly, MtPRR9a, 9b, 7, and 5 formed complexes with MtTPL/MtTPR. Combining the transcriptional regulatory relationship between *MtPRR* and *MtGI* or *MtLHY* in clock regulation, we put forward a plausible hypothesis that MtPRR9, 7, and 5 may recruit the MtTPL/MtTPRs to form complexes to suppress the expression of *MtLHY* and *MtGI* to regulate plant development (Figure 7). 

Modern agriculture needs to precisely control the flowering time against the emerging patterns of climate change [58]. Simultaneous disruption of *MtPRR9a*, *9b*, *7*, and *5* induced early flowering in *M. truncatula*, providing genetic evidence for the function of the circadian clock in *PRRs* flowering regulation. This result is consistent with the previously described role of *PRRs* as negative flowering time regulators in rice, sorghum, and soybean, but is inconsistent with the function of *PRRs* in *Arabidopsis* [19,34,39,41,43]. Importantly, the flowering time in *M. truncatula* is advanced in the long-day and prolonged cold conditions, so the negative function of *MtPRRs* in flowering time regulation may be helpful to improve the adaptability to high latitudes and increase the yield of the leguminous forage. According to the reports above, the circadian clock *PRRs* may be positive regulators in flowering time regulation in long-day plants, and negative regulators in short-day plants. However, the *PRRs* negatively control the flowering time in the long-day species *Medicago truncatula*, indicating a special regulatory mechanism. In *Arabidopsis*, the *GI*-*CONSTANS* (*CO*)-*FT* module is the major photoperiod pathway for floral regulation, and PRRs can regulate the transcription of *CO* or stabilize the CO protein to enhance the transcription of FT in promoting flowering time [59]. In rice, the photoperiod pathway is conserved, but the *OsPRRs* inhibit the expression of *OsFT* under long-day conditions [60]; similarly, *SbPRR37* also represses the expression *SbFT* in sorghum [39]. These results indicate that the clock *PRRs* function divergently in the long-day and short-day plants, and the flowering regulation appears to act in a *CO*-*FT* dependent manner. However, three *CO-like* genes, named *MtCOLa*, *MtCOLb*, and *MtCOLc*, are not involved in the photoperiodic flowering in *M. truncatula* [61]. Furthermore, overexpression of the *MtCDF* (*CYCLING DOF FACTOR*) gene results in delayed flowering time, and the expression levels of the *MtCOL* genes are unchanged [62]. Moreover, *FTa1* plays a major role in flowering time regulation but is not involved in the photoperiodic induction of flowering [29]. Based on these results, we assume that *MtPRRs* regulating flowering time in *M. truncatula* is independent of the *CO*-*FT* module, and they may regulate flower by repressing the *MtGI* and *MtLHY* at a transcriptional level (Figure 7). Their orthologs are required for normal circadian rhythms and photoperiodic flowering in *Arabidopsis*, and *LHY* also represses the *GI* expression [63,64]. In conclusion, studying the functions of the *PRR* genes will be helpful to understand the mechanism of circadian rhythm in *Medicago truncatula*, and provides useful information to improve the agricultural traits by the circadian clock genes.

## 4. Materials and Methods

### 4.1. Plant Materials and Growth Conditions

*Medicago truncatula* ecotype R108 was used as the wild-type accession for all the experiments described in this study. Mutants of *mtprr9a-1*, *mtprr9b-1*, *mtprr7-1*, and *mtprr5-1* were identified from the *Tnt1* retrotransposon-tagged mutant collection of *M. truncatula* [65]. For flowering time measurement, the plants were grown in the greenhouse at 22 °C, with a long-day (16 h light and 8 h dark) photoperiod with a light intensity of 150 μmol/m^−2^/s^−1^ and 70–80% relative humidity. For the circadian clock genes expression analyses, plants were grown at 22 °C in a light incubator with 70 and 80% relative humidity, and a light intensity of 90 µmol/m^−2^/s^−1^. Plants were grown under a long-day photoperiod for three weeks, and then transferred to a short-day photoperiod (12 h light and 12 h dark) for one week, followed by two days of constant light for sampling. For diurnal rhythmic analyses of *MtPRRs*, the wild type was grown in a short-day period for four weeks, and the leaves were collected for further analysis.

### 4.2. Identification and Phylogenetic Analysis of MtPRRs

The sequences of the AtPRRs were obtained from the Arabidopsis Information Resource (TAIR) database (http://www.arabidopsis.org/, accessed on 1 March 2020). AtPRR proteins were used to perform a BLASTP search against the sequence database of the *Medicago truncatula* in Phytozome (https://phytozome-next.jgi.doe.gov/, accessed on 20 March 2020). 

To study the phylogenetic relationships between PRRs in *M. truncatula* and *Arabidopsis*, seven identified MtPRR proteins in *M. truncatula* and five *Arabidopsis* PRR proteins were used to generate the phylogenetic tree. Multiple sequence alignments were executed using CLUSTALW online (http://www.genome.jp/tools-bin/clustalw, accessed on 27 March 2020). Then, the phylogenetic tree was generated with the neighbor-joining method and 1000 bootstrap replications using the MEGA7.1 program in the p-distance model. The scale bar was shown to indicate the genetic distance based on branch length.

### 4.3. Conserved Domains Analysis

The PRR proteins were aligned using Clustal X2, and the GeneDoc 2.7 software was used for homology shading [66,67].

### 4.4. Subcellular Localization Analysis

To generate the constructs, the full-length *MtPRR9a*, *MtPRR9b*, *MtPRR7*, and *MtPRR5* coding sequences (CDS) were amplified from the wild type and cloned into the pENTR/D-TOPO cloning vector (Invitrogen, Carlsbad, CA, USA), then recombined with destination vector pEarleyGate 103, using the Gateway LR recombination reactions (Invitrogen, Carlsbad, CA, USA). To observe the subcellular localization, the *35S:MtPRR9a-GFP*, *35S:MtPRR9b-GFP*, *35S:MtPRR7-GFP*, *35S:MtPRR5-GFP*, and the empty pEarleygate 103 were used and transformed into tobacco leaves for analyses. The infiltrated leaves were incubated for 48 h, and the GFP signal was observed under confocal microscopy (Zeiss, Jena, Germany). 

### 4.5. RNA Extraction, RT-PCR, qRT-PCR, and Statistical Analysis

The leaves and the other tissues were collected for total RNA isolation using the Trizol-RT Reagent (Invitrogen). For RT-PCR analysis, RNA was extracted from vegetative buds of wild type and mutant lines. RNA extraction, cDNA synthesis, RT-PCR, and qRT-PCR analyses were performed as described previously [46]. The primers used for RT-PCR and RT-qPCR analysis are listed in Appendix A. The *t*-test was used to compare the means of different populations.

### 4.6. Double/Triple/Quadruple Mutant Generation

To obtain the *mtprr9a-1 mtprr9b-1* double mutant, the *mtprr9a-1* and *mtprr9b-1* homozygous plants were used as parents and crossed with each other to generate F1 plants. To obtain the *mtprr9a-1 mtprr9b-1 mtprr5-1* triple mutant, the *mtprr9a-1 mtprr9b-1* and *mtprr5-1* were used as parents for crossing. To generate *mtprr9a-1 mtprr9b-1 mtprr5-1 mtprr7-1* quadruple mutant, the *mtprr9a-1 mtprr9b-1 mtprr5-1* and *mtprr7-1* were crossed. The F1 plants were identified by PCR, and the double, triple, and quadruple mutants were identified by PCR in the F2 segregating population. 

### 4.7. Y2H and BiFC Assays

The Y2H assay was performed using the Matchmaker Gold System (Clontech, Shiga, Japan). To detect the interactions between MtTPL and MtPRRs, the CDS of *MtTPL* were cloned into the pGADT7 (AD) or pGBKT7 (BD) vector, and the CDS of *MtPRRs* were cloned into the pGBKT7 or pGADT7 vector, correspondingly. To detect the interactions between MtTPRs and MtPRR9a, the CDS of *MtTPRs* were cloned into pGADT7, while the *MtPRR9a* CDS was cloned into pGBKT7. To check the interactions among MtPRRs, the CDS of *MtPRRs* were cloned and introduced into pGADT7 or pGBKT7. Protein–protein interactions were performed as described previously according to the manufacturer’s protocol [68]. 

For the BiFC assays, *MtPRR9a* was cloned to the pEarleyGate202-YC vector, while *MtPRR9b*, *MtPRR7*, and *MtPRR5* were cloned into pEarleyGate201-YN using the Gateway system (Invitrogen). All the constructs were introduced into the *Agrobacterium tumefaciens* EHA105 strain. Various combinations of transformed cells were simultaneously infiltrated into tobacco leaves. After infiltration for 48 h, the yellow fluorescent protein (YFP) signals were observed under a confocal laser scanning microscope (Zeiss).

### 4.8. Accession Numbers

Accession numbers for the genes in this article are as follows: MtPRR9a: Medtr7g118260; MtPRR9b: Medtr8g024260; MtPRR7: Medtr1g067110; MtPRR5: Medtr3g092780; MtPRR3: Medtr4g061360; MtTOC1a: Medtr3g037390; MtTOC1b: Medtr4g108880; MtGI: Medtr1g098160; MtLHY: Medtr7g118330; MtTPL: Medtr4g009840; MtTPR1: Medtr2g104140; MtTPR2: Medtr4g120900; MtTPR3: Medtr1g083700; MtTPR4: Medtr7g112460; MtTPR5: Medtr4g114980.

## 5. Conclusions

In this study, we identified the *PRR* genes in *M. truncatula*. The *MtPRR9a*, *9b*, *7*, and *5* gene expression profiles were characterized in different tissues, and the proteins were located in the nucleus. The diurnal expression of *MtPRR9a*, *9b*, *7*, and *5* also showed a clear circadian rhythm, suggesting that they are important components of the circadian clock. Further investigations showed that *MtPRR9a*, *9b*, *7*, and *5* function redundantly in flowering time. At last, MtPRR9a, 9b, 7, 5 could form heterodimers with MtTPL/MtTPRs or with each other. Further study is also needed to elucidate the functions of all the *MtPRRs* that are involved in the circadian rhythm and plant development.

## Figures and Tables

**Figure 1 ijms-24-16834-f001:**
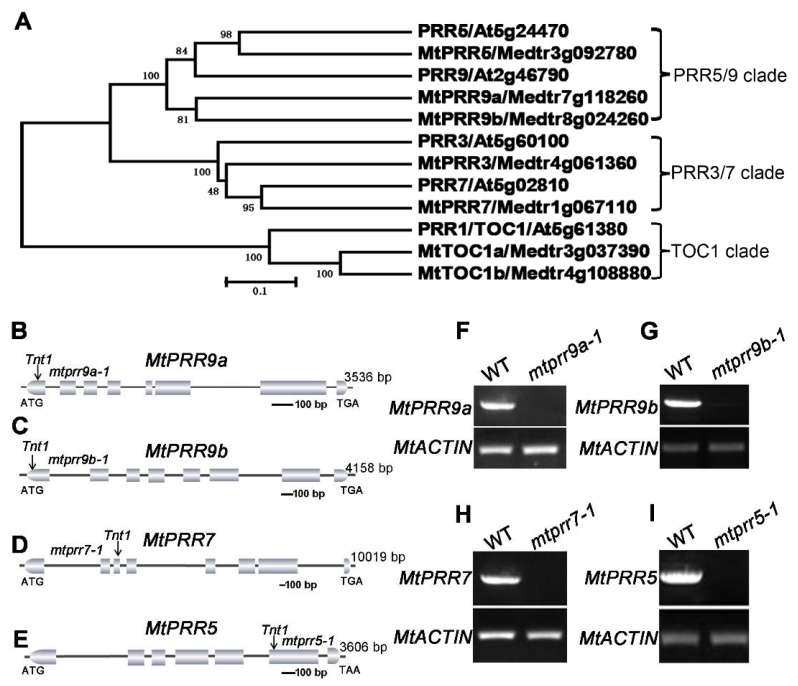
Molecular characterization of *MtPRR9a*, *9b*, *7*, and *5* in *M. truncatula*. (**A**) Phylogenetic tree analysis of MtPRR putative orthologs in *Arabidopsis* and *M. truncatula*. The PRR5/9, PRR3/7, and TOC1 clade are shown. The scale bar indicates the genetic distance based on branch length. (**B**–**E**) Schematic representations of the gene structures of *MtPRR9a*, *9b*, *7*, *5* showing the *Tnt1* insertion sites in relative mutants. The positions of the ATG start and TAA/TGA stop codons are shown. Vertical arrows mark the location of *Tnt1* retrotransposons in mutants. Introns are represented by lines and exons are represented by boxes. (**F**–**I**) RT-PCR shows the transcript abundance of *MtPRR9a*, *9b*, *7*, and *5* in the leaf of wild type and relative mutants. *MtACTIN* was used as the control.

**Figure 2 ijms-24-16834-f002:**
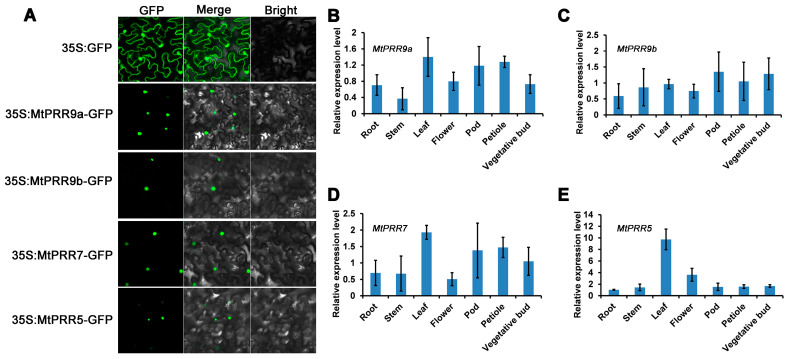
Subcellular localization and expression patterns of *MtPRR9a*, *9b*, *7*, *5* in *M. truncatula*. (**A**) The subcellular localization of MtPRR-GFP fusion proteins. Free GFP was used as the control. (**B**–**E**) The expression levels of *MtPRR9a*, *9b*, *7*, and *5* in different organs. *MtUBIQUITIN* was used as the internal control. Values are shown as means ± SD of three biological replicates.

**Figure 3 ijms-24-16834-f003:**
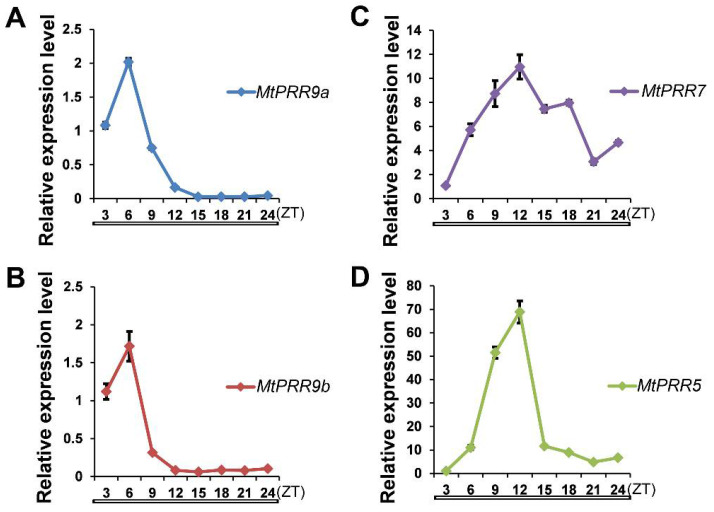
*MtPRR9a* (**A**), *9b* (**B**), *7* (**C**), and *5* (**D**) transcript levels in the long-day conditions of the 4-week-old wild type plant. *MtUBIQUITIN* was used as the internal control. Values are shown as means ± SD of three biological replicates. White bars at the bottom indicate periods of constant light, and ZT means the zeitgeber time.

**Figure 4 ijms-24-16834-f004:**
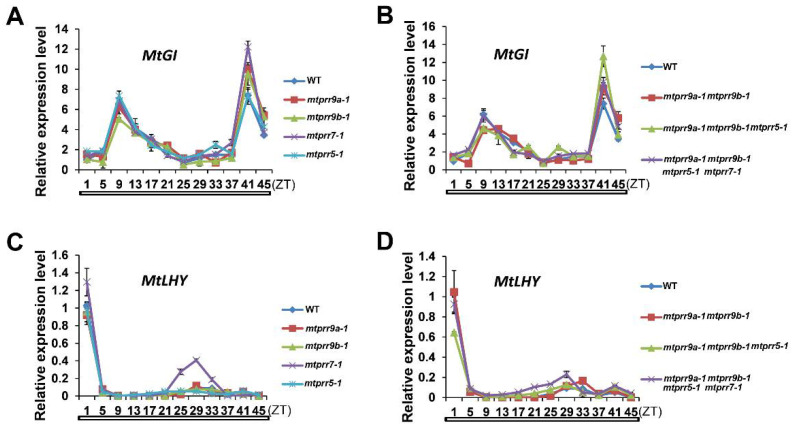
Regulation of *MtPRR9a*, *9b*, *7*, *5* on the circadian expression of clock genes in continuous light. The transcriptional behaviors of *MtGI* (**A**,**B**) and *MtLHY* (**C**,**D**) in the leaves of 4-week-old wild type and *mtprr* single, double, triple, and quadruple mutants grown under constant light conditions. Values are shown as means ± SD of three biological replicates. White bars at the bottom indicate periods of constant light, and ZT means the zeitgeber time.

**Figure 5 ijms-24-16834-f005:**
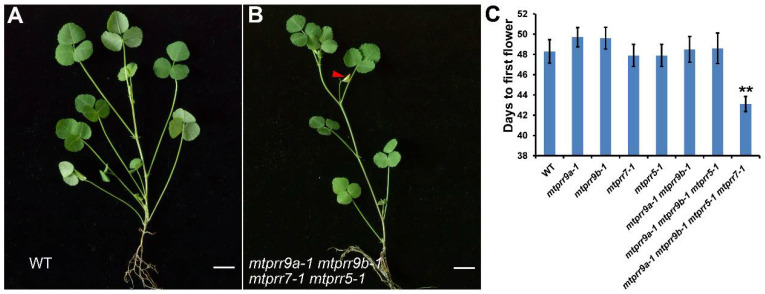
Early flowering phenotype in the *mtprr9a-1 mtprr9b-1 mtprr7-1 mtprr5-1* plants. (**A**,**B**) The 44-day-old plants of the wild type (**A**) and *mtprr9a-1 mtprr9b-1 mtprr7-1 mtprr5-1* mutant (**B**) in LDs. Arrows indicate the flowers of the mutant. Bars = 1 cm. (**C**) The days to the first flower of wild type and relative mutants in the LDs. Values are shown as means ± SD (*n* = 20). **: means differ significantly (*p <* 0.01).

**Figure 6 ijms-24-16834-f006:**
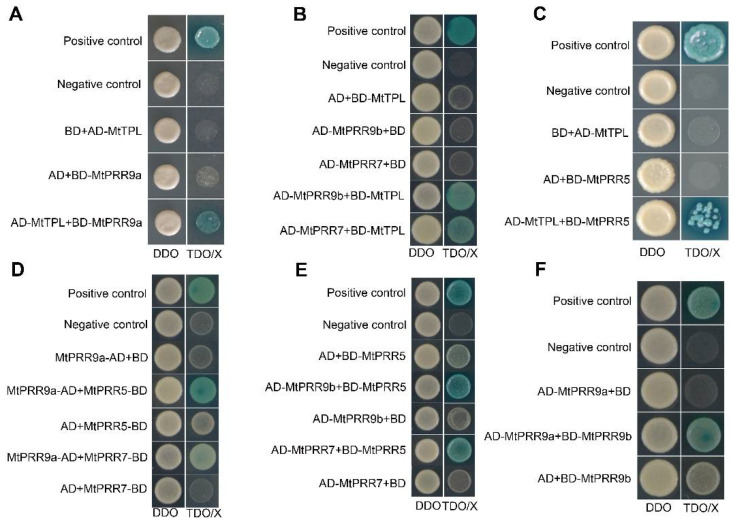
Identification of the interactions between MtTPL and MtPRR9a, 9b, 7, and 5, and interactions among MtPRR9a, 9b, 7, and 5 by Y2H. (**A**–**C**) Validation of MtPRR9a, 9b, 7, and 5 proteins interacting with MtTPL. (**D**–**F**) Validation of MtPRR9a protein interacting with MtPRR5, 7, and 9b, and MtPRR5 protein interacting with MtPRR7 and 9b. All transformants can grow on DDO (SD/-Leu/-Trp as double dropout) medium. Yeast colonies that were able to grow on TDO (SD/-His/-Leu/-Trp) with X-α-Gal and displayed blue coloration confirmed the protein–protein interaction.

**Figure 7 ijms-24-16834-f007:**
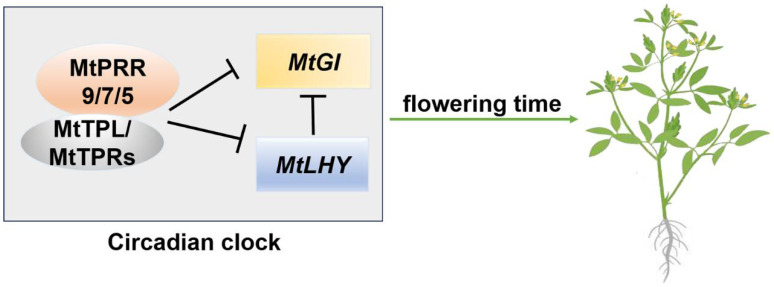
Model for the proposed roles of *MtPRR9*, *7*, and *5* in *M. truncatula*. The complexes between MtPRR9, 7, and 5 and MtTPL/MtTPRs are responsible for the downregulation of *MtLHY* and *MtGI*. The MtPRR9-, 7-, and 5-associated circadian clock is involved in the flowering time regulation.

## Data Availability

The data presented in this study are available in Appendix A here.

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
