# Peer review of "The Roles of the *PSEUDO-RESPONSE REGULATORs* in Circadian Clock and Flowering Time in *Medicago truncatula"

_ijms, 2023, doi:10.3390/ijms242316834_

Round 1

Reviewer 1 Report

Comments and Suggestions for Authors

Wang and colleagues presented a very interesting study on the roles of PRRs in the regulation of the circadian clock and of flowering time in Mt. I think this study is well done and well presented. The methodologies used are solid and well support the authors’ claims. The suppressive role of PRRs on the regulation of LHY and GI clearly emerges thanks to the use of mutants which is also reflected in a negative role on the regulation of the flowering times. The use of the quadruple mutant allows to clarify this specific function very well. Preliminary analyses regarding protein localization by GFP and in silico analyses of conserved motifs also highlight the potential role of PRRs as transcription factors. The study on protein interactions is also interesting. For all these reasons I consider the manuscript absolutely publishable as I believe it can be very useful to those who study these aspects in plants.

Author Response

RESPONSE: Thank you for your comment.

Reviewer 2 Report

Comments and Suggestions for Authors

The paper describes that charactrtization of PSEUDO-RESPONSE REGULATORs (PRRs) genes in legume model plant, Medicago truncatula.  PRRs play key roles in the circadian rhythms and flowering in plants.  There are many of information on PRRs in Arabidopsis thaliana.  However, there is a few reports on PRRs in legume plants.  Therefore, the paper includes new findings and provides valuable information on improvement of plant breeding programs.

My suggestion points are as follows:

Discussion is too short and is not sufficiently well described.  I think it would be good to include a few considerations, e.g. differences from Arabidopsis, especially in comparison to Poaceae species such as rice and sorghum in term of orthologs and evolutional aspect. Also, regulation function and interaction of other flowering genes and so on.

Aastract   ‘Tnt1 retransposon-tagged’ should added.

L45  ptt7  -->  prr7

L57-  Description on PRR gene should be moved to the beginning of the Introduction.

Author Response

1: Discussion is too short and is not sufficiently well described.  I think it would be good to include a few considerations, e.g. differences from Arabidopsis, especially in comparison to Poaceae species such as rice and sorghum in term of orthologs and evolutional aspect. Also, regulation function and interaction of other flowering genes and so on.

RESPONSE: Thank you for your suggestion. We have revised the Discussion section as follows:

“According to the previous study, there are five members of clock PRR genes in Arabidopsis thaliana, Sorghum bicolour and Oryza sativa, all including one TOC1/PRR1 member gene, two genes in the PRR5/9 clade and two genes in the PRR3/7 clade [41, 56]. However, seven PRR genes were shown in the Medicago truncatula genome, including two genes in the TOC1/PRR1 clade, three members in the PRR5/9 clade, and two PRR3/7 members. This is mainly because that PRRs were expanded in M. truncatula during evolution, and the duplication event tend to happen in the TOC1/PRR1 clade and PRR5/9 clade, which is also seen in the PRR family of Populus trichocarpa and Vitis vinifera [56]. Like the PRR proteins in Arabidopsis, rice, sorghum and other plants, the PRRs in M. truncatula are also nuclear-localized with an N-terminal PR domain followed by the C-terminal CCT motif, the structure and location similarity imply their potential functional conservation in monocots and dicots species.”

“Importantly, the flowering time in M. truncatula is advanced in the long-day and prolonged cold conditions, so the negative function of MtPRRs in flowering time regulation may be helpful to improve the adaptability to high latitudes and increase the yield of the leguminous forage.”

“However, the PRRs negatively control the flowering time in the long-day species Medicago truncatula, indicating a special regulatory mechanism. In Arabidopsis, the GI-CONSTANS (CO)-FT module is the major photoperiod pathway for floral regulation, PRRs can regulate the transcription of CO or stabilize the CO protein to enhance the transcription of FT in promoting flowering time [59]. In rice, the photoperiod pathway is conserved, but the OsPRRs inhibit the expression of OsFT under long-day conditions [60], similarly, SbPRR37 also represses the expression SbFT in sorghum [39]. These results indicate that the clock PRRs function divergently in the long-day and short-day plants, and the flowering regulation appears to act in a CO-FT dependent manner. However, three CO-like genes, named MtCOLa, MtCOLb and MtCOLc, are not involved in the photoperiodic flowering in M. truncatula [61]. Besides, overexpression of the MtCDF (CYCLING DOF FACTOR) gene exhibits delayed flowering time, and the expression levels of the MtCOL genes are unchanged [62]. Moreover, FTa1 plays a major role in flowering time regulation, but is not involved in the photoperiodic induction of flowering [29]. Based on these, we assume that MtPRRs regulating flowering time in M. truncatula is independent of the CO-FT module, and they may regulate flower by repressing the MtGI and MtLHY at transcriptional level (Figure 7).”

2: Aastract   ‘Tnt1 retransposon-tagged’ should added.

 RESPONSE: Thank you for pointing this out. We have revised in the Abstract section (lines 13-14) as follows:

“and isolated their Tnt1 retrotransposon-tagged mutants.”

3: L45  ptt7  -->  prr7

RESPONSE: Thank you for pointing this out. We have revised it in line 47.

prr5 prr7 prr9 triple mutant”

4: L57-  Description on PRR gene should be moved to the beginning of the Introduction.

RESPONSE: Thank you for pointing this out. We have revised it in the Introduction section (lines 41-42).

“The PRR proteins have an N-terminal Pseudo-Receiver (PR) domain and a C-terminal CTT (CONSTANS, CONSTANS-LIKE, TOC1) domain”.

Reviewer 3 Report

Comments and Suggestions for Authors

Review of the Paper: " The Roles of the PSEUDO-RESPONSE REGULATORs in Circadian Clock and Flowering Time in Medicago truncatula"

 Objective and Scope of the Study:

The introduction of the paper lacks sufficient information concerning legume plants, and the main research question regarding the adaptation of "Medicago truncatula" is merely mentioned. There is a lack of connection between the research objective and the potential application of the obtained results. The paper is currently considered predominantly technical; however, it is recommended to supplement the research objective to better integrate the acquired findings within the context of legume plant adaptation to higher latitudes.

 Originality and Significance of the Topic:

The research topic is deemed significant in the field, especially in the context of climate change and the adaptation of legume plants to the variable diurnal sunlight cycle. The paper distinguishes itself with a methodologically sound execution of the experimental section, although there is a need for a more explicit demonstration of the application of the obtained results.

 Conclusions in Comparison with Other Publications:

The paper makes a significant contribution to the thematic area by emphasizing issues related to climate change and the adaptation of legume plants. However, the reference to other published works is not sufficiently explored, leaving room for further investigation.

 Improvements in Methodology:

Authors should consider expanding information on distance measures and agglomeration methods, especially concerning the presentation in Figure 1. The absence of these details may impose limitations on the understanding and reproducibility of the experiments.

 Consistency of Conclusions with Evidence:

The conclusions presented in the paper are consistent with the provided evidence and arguments. The authors successfully address the main research question.

 Appropriate References:

The references in the paper are pertinent to the topic and provide solid theoretical support for the presented results.

 Additional Comments on Tables and Figures:

The tables and figures in the paper are appropriate, accurate, and effectively represent the research results. There is no need for additional comments in this area.

In summary, the paper constitutes a valuable contribution to the field of legume plant adaptation to climate change. However, it is recommended to improve the contextualization of the research objective and expand methodological information.

Author Response

1: Objective and Scope of the Study:

The introduction of the paper lacks sufficient information concerning legume plants, and the main research question regarding the adaptation of "Medicago truncatula" is merely mentioned. There is a lack of connection between the research objective and the potential application of the obtained results. The paper is currently considered predominantly technical; however, it is recommended to supplement the research objective to better integrate the acquired findings within the context of legume plant adaptation to higher latitudes.

RESPONSE: Thank you for pointing this out. We have revised it in the Introduction and Discussion sections as follows:

Medicago truncatula, functions as a model plant in legume forage, is a temperate species. The flowering time of M. truncatula is promoted by exposure to a prolonged period of cold conditions, so it is cultivated in a relatively limited geographic range [27, 28]. Exploring the genes involved in flowering time regulation will be helpful to understand the climate adaptation of M. truncatula, and some important genes have been reported so far. For example, FTa1, one of the FLOWERING LOCUS T (FT) orthologs in M. truncatula, is a key regulator of flowering time, and the loss of function of FTa1 exhibits late flowering while overexpression of MtFTa1 accelerates flowering time [29]. A legume-specific gene in Glycine max was named E1, and the mutation of the homolog of E1 causes a delay in flowering in M. truncatula [30]. In addition, SUPPRESSOR OF OVEREXPRESSION OF CONSTANS1 (SOC1), PHYTOCHROME A and FRUITFULL-LIKE genes promote the flowering in M. truncatula [31-33]. But the functions of MtPRRs in flowering time regulation are still unknown.”

“What’s more, Time of Flowering 11 (Tof11) and Tof12, two homologous genes of AtPRR3 in Glycine max, act via GmLHY to promote the transcription of E1 and delay flowering time under the long-day conditions, and improve the adaptability of soybean to high latitudes during domestication [42-44].”

“Importantly, the flowering time in M. truncatula is advanced in the long-day and prolonged cold conditions, so the negative function of MtPRRs in flowering time regulation may be helpful to improve the adaptability to high latitudes and increase the yield of the leguminous forage.”

2: Originality and Significance of the Topic:

The research topic is deemed significant in the field, especially in the context of climate change and the adaptation of legume plants to the variable diurnal sunlight cycle. The paper distinguishes itself with a methodologically sound execution of the experimental section, although there is a need for a more explicit demonstration of the application of the obtained results.

RESPONSE: Thank you for your comment.

3: Conclusions in Comparison with Other Publications:

The paper makes a significant contribution to the thematic area by emphasizing issues related to climate change and the adaptation of legume plants. However, the reference to other published works is not sufficiently explored, leaving room for further investigation.

RESPONSE: Thank you for pointing this out. We have added the published references in the Introduction section as follows:

Medicago truncatula, functions as a model plant in legume forage, is a temperate species. The flowering time of M. truncatula is promoted by exposure to a prolonged period of cold conditions, so it is cultivated in a relatively limited geographic range [27, 28]. Exploring the genes involved in flowering time regulation will be helpful to understand the climate adaptation of M. truncatula, and some important genes have been reported so far. For example, FTa1, one of the FLOWERING LOCUS T (FT) orthologs in M. truncatula, is a key regulator of flowering time, and the loss of function of FTa1 exhibits late flowering while overexpression of MtFTa1 accelerates flowering time [29]. A legume-specific gene in Glycine max was named E1, and the mutation of the homolog of E1 causes a delay in flowering in M. truncatula [30]. In addition, SUPPRESSOR OF OVEREXPRESSION OF CONSTANS1 (SOC1), PHYTOCHROME A and FRUITFULL-LIKE genes promote the flowering in M. truncatula [31-33]. But the functions of MtPRRs in flowering time regulation are still unknown.”

“What’s more, Time of Flowering 11 (Tof11) and Tof12, two homologous genes of AtPRR3 in Glycine max, act via GmLHY to promote the transcription of E1 and delay flowering time under the long-day conditions, and improve the adaptability of soybean to high latitudes during domestication [42-44].”

4: Improvements in Methodology:

Authors should consider expanding information on distance measures and agglomeration methods, especially concerning the presentation in Figure 1. The absence of these details may impose limitations on the understanding and reproducibility of the experiments.

RESPONSE: Thank you for pointing this out.

(1) We completed the genes information of MtPRR9a, 9b, 7 and 5 in Figure 1B-E.

(2) We revised the phylogenetic analysis method in lines 343-349 as follows:

“To study the phylogenetic relationships between PRRs in M. truncatula and Arabidopsis, seven identified MtPRR proteins in M. truncatula and five Arabidopsis PRR proteins were used to generate the phylogenetic tree. Multiple sequence alignments were executed using CLUSTALW online (http://www.genome.jp/tools-bin/clustalw). Then, the phylogenetic tree was generated with the Neighbor-Joining method and 1000 Bootstrap Replications using the MEGA7.1 program in the p-distance model. The scale bar was shown to indicate the genetic distance based on branch length.”

5: Consistency of Conclusions with Evidence:

The conclusions presented in the paper are consistent with the provided evidence and arguments. The authors successfully address the main research question.

RESPONSE: Thank you for your comment.

6: Appropriate References:

The references in the paper are pertinent to the topic and provide solid theoretical support for the presented results.

RESPONSE: Thank you for your comment.

 7: Additional Comments on Tables and Figures:

The tables and figures in the paper are appropriate, accurate, and effectively represent the research results. There is no need for additional comments in this area.

In summary, the paper constitutes a valuable contribution to the field of legume plant adaptation to climate change. However, it is recommended to improve the contextualization of the research objective and expand methodological information.

RESPONSE: Thank you for your comment.